# Diagnostic and Prognostic Roles of CDX2 Immunohistochemical Expression in Colorectal Cancers

**DOI:** 10.3390/diagnostics12030757

**Published:** 2022-03-20

**Authors:** Hong Bae Choi, Jung-Soo Pyo, Soomin Son, Kyungdoc Kim, Guhyun Kang

**Affiliations:** 1Department of Surgery, Daehang Hospital, Seoul 06699, Korea; cozypoppy@gmail.com; 2Department of Pathology, Uijeongbu Eulji Medical Center, Eulji University School of Medicine, Uijeongbu-si 11759, Korea; jspyo@eulji.ac.kr; 3Division of Molecular Life and Chemical Sciences, College of Natural Sciences, Ewha Woman’s University, Seoul 03760, Korea; smsonaj00@gmail.com; 4VUNO Inc., Seoul 06541, Korea; kyungdoc.kim@vuno.co; 5Department of Pathology, Daehang Hospital, Seoul 06699, Korea

**Keywords:** colorectal cancer, CDX2, immunohistochemistry, diagnosis, prognosis, meta-analysis

## Abstract

The study is aimed to evaluate the diagnostic and prognostic role of the immunohistochemical expression of the Caudal-type homeobox transcription factor 2 (CDX2) in colorectal cancers (CRCs) through a meta-analysis. By searching relevant databases, 38 articles were eligible to be included in this study. We extracted the information for CDX2 expression rates and the correlation between CDX2 expression and clinicopathological characteristics. The estimated rates of CDX2 expression were 0.882 [95% confidence interval (CI) 0.774–0.861] and 0.893 (95% CI 0.820–0.938) in primary and metastatic CRCs, respectively. Furthermore, based on their histologic subtype, CDX2 expression rates of adenocarcinoma and medullary carcinoma were 0.886 (95% CI 0.837–0.923) and 0.436 (95% CI 0.269–0.618), respectively. There was a significant difference in CDX2 expression rates between adenocarcinoma and medullary carcinoma in the meta-regression test (*p* < 0.001). In addition, CDX2 expression was significantly lower in CRCs with the BRAF^V600E^ mutation than in CRCs without mutation. Patients with CDX2 expression had better overall and cancer-specific survival rates than those without CDX2 expression. Thus, CDX2 is a useful diagnostic and prognostic marker CRCs.

## 1. Introduction

The Caudal-type homeobox transcription factor 2 (CDX2) gene is a specific intestinal transcription factor expressed in the nuclei of intestinal epithelial cells [1,2]. The CDX2 gene is involved in the embryonic development and differentiation of the intestine [1]. Because CDX2 gene transcription is involved in the colon and small intestine cells in humans, it can be used to differentiate from that of other origins. Traditionally, immunohistochemical staining for cytokeratin 7 and 20 has been the most widely used marker in various adenocarcinomas, including colorectal cancers (CRCs) [2]. The most common expression pattern of CRCs is cytokeratin 20 positive and cytokeratin 7 negative [3,4,5]. Typically, CDX2 immunohistochemistry is considered useful as a single marker. Because CDX2 plays a role in cell proliferation and differentiation [6,7]. The downregulation of CDX2 expression may be associated with loss of differentiation [7]. 

Previous studies have reported that a loss of CDX2 expression is correlated with poor survival [8,9,10]. Thus, the prognostic implications of CDX2 expression should be considered. Interestingly, CDX2 can be expressed in other malignant tumors, such as lung, ovarian, biliary, and urinary bladder carcinomas [11]. In CRCs, CDX2 expression rates range from 26.7% to 100.0% [2,8,10,12,13,14,15,16,17,18,19,20,21,22,23,24,25,26,27,28,29,30,31,32,33,34,35,36,37,38,39,40,41,42,43]. Even though CDX2 is a specific marker, it is not positive in all cases and its expression can be affected by various factors, such as tumor type and evaluation methods.

Although immunohistochemistry is used in many pathology laboratories, stain methods and interpretation of results can vary between laboratories. The aim of this study is to evaluate the diagnostic and prognostic roles of CDX2 expression in CRC through a meta-analysis. In addition, the authors perform a detailed analysis on the histologic subtypes and evaluation criteria of CRCs.

## 2. Materials and Methods

### 2.1. Literature Search and Selection Criteria

Relevant articles were obtained by searching the PubMed and MEDLINE databases through 30 September 2021. The search was performed using ‘CDX2 AND colon ANDimmunohistochemistry’ as search terms. The titles and abstracts of all returned articles were screened for exclusion. Review articles were also screened to find additional eligible studies. English language studies addressing CDX2 immunohistochemistry (IHC) expression in human CRC and correlation between CDX2 IHC expression and clinicopathological characteristics were included. Case reports or review articles were excluded. Finally, 38 reports were included for the meta-analysis (Figure 1).

### 2.2. Data Extraction

The following information was collected and verified from the full texts of eligible studies [2,8,10,12,13,14,15,16,17,18,19,20,21,22,23,24,25,26,27,28,29,30,31,32,33,34,35,36,37,38,39,40,41,42,43]: first author’s name, publication date, study location, number of patients analyzed, antibody manufacturer, dilution ratio, cut-offs for assessing CDX2 IHC expression, and tumor type. In addition, the correlations between CDX2 expression and clinicopathological characteristics and survivals. Any disagreements were resolved by consensus.

### 2.3. Statistical Analyses

To perform the meta-analysis, we used the Comprehensive Meta-Analysis software package (Biostat, Englewood, NJ, USA). The CDX2 IHC expressions were investigated from eligible studies. Subgroup analysis based on histologic subtype was performed. In addition, the estimated CDX2 expressions of primary and metastatic CRCs were compared. The correlations between CDX2 expression and clinicopathological characteristics were evaluated. Correlations between claudin-1 IHC expression and survival were measured by hazard ratio (HR) obtained from the eligible study data. To obtain survival information, we aggregated the estimated HR and its standard error using given parameters, which were the HR point estimate, log-rank statistic or its *p*-value, O–E statistic (difference between numbers of observed and expected events), or its variance [44]. If the extractable data only included the survival curve, two persons independently extracted survival rates to reduce reading variability according to Parmar’s recommendation [44]. The meta-analysis was performed by fixed-effects and random-effects models. The values pooled using the random-effects models were utilized for interpretation. Subsequently, a study showing results of an estimated HR <1 without a 95% confidence interval (CI) overlapping 1 implied better survival. Because eligible studies used various antibodies and evaluation criteria for various populations, random-effects models were more suitable than fixed-effects models. Heterogeneous and sensitivity analyses were conducted to assess the heterogeneity of eligible studies and the impact of each study on the combined effect, respectively. Heterogeneity between studies was checked by the Q and I2 statistics and demonstrated *p*-values. For assessment of publication bias, Begg’s funnel plot and Egger’s test were performed. The results were considered statistically significant when *p* < 0.05.

## 3. Results

### 3.1. Selection and Characteristics of Studies

Two hundred four reports were identified in the database search. Detailed information of each eligible study, including antibody manufacturer and evaluation criteria, is shown in Table 1. Among excluded articles, 57 were excluded due to the results of other diseases. Other studies were excluded because they lack sufficient information (n = 44), used animals or cell lines (n = 31), non-original articles (n = 29), or were non-English (n = 5).

### 3.2. CDX2 Expression Rates in Primary and Metastatic Colorectal Carcinoma

CDX2 expression rate of primary and metastatic CRCs was 0.882 (95% CI 0.774–0.861) and 0.893 (95% CI 0.820–0.938), respectively (Table 2). In the subgroup analysis based on histologic subtypes, estimated CDX2 expression rates were 0.886 (95% CI 0.837–0.923), 0.882 (95% CI 0.632–0.970), 0.436 (95% CI 0.269–0.618), 0.873 (95% CI 0.756–0.938), and 0.772 (95% CI 0.46–0.944) in adenocarcinoma, mucinous carcinoma, medullary carcinoma, micropapillary carcinoma, and signet ring carcinoma, respectively. CDX2 expression rate of medullary carcinoma was significantly lower than other histologic subtypes in the meta–regression test (*p* < 0.001). In metastatic CRCs, CDX2 expression rates were 0.984 (95% CI 0.789–0.999), 0.896 (95% CI 0.605–0.980), 0.962 (95% CI 0.597–0.998), and 0.967 (95% CI 0.634–0.998) in lung, ovary, urinary bladder, and uterine cervix, respectively.

Next, we analyzed the effect of evaluation criteria for CDX2 expression. In primary CRCs, CDX2 expression rates were 0.817 (95% CI 0.731–0.880), 0.816 (95% CI 0.713–0.888), 0.839 (95% CI 0.699–0.922), 0.702 (95% CI 0.173–0.963), and 0.875 (95% CI 0.786–0.930) in criteria 0%, 5%, 10%, 25%, and 50% subgroups, respectively. In metastatic CRCs, CDX2 expression rates of criteria 0%, 1%, 10%, and 50% subgroups were 0.942 (95% CI 0.877–0.974), 0.969 (95% CI 0.650–0.998), 0.848 (95% CI 0.687–0.934), and 0.855 (95% CI 0.569–0.963), respectively. However, there were no significant differences of CDX2 expression rates between evaluation criteria in primary and metastatic CRCs in the meta–regression test. In medullary carcinoma subgroup, a significant difference of CDX2 expression rates between cut-offs was found. CDX2 expression rates were 0.601 (95% CI 0.502–0.694), 0.400 (95% CI 0.192–0.652), and 0.188 (95% CI 0.062–0.447) in criteria 0%, 10%, and 25%, respectively.

### 3.3. Correlation between CDX2 Expression and Clinicopathological Characteristics

We compared the CDX2 expression rates based on various clinicopathologic parameters. CDX2 expression rates were not significantly different according to sex, tumor location, histologic grade, pT stage, lymph node metastasis, venous, lymphatic, perineural invasion, and pTNM stage in the meta-regression test (Table 3). In addition, there was no significant difference between CRC with and without PD-L1 expression (*p* = 0.246 in the meta-regression test).

Next, the correlations between CDX2 expression and genetic mutation status were evaluated (Table 4). CRCs with BRAF mutation showed significantly lower CDX2 expression than those without BRAF mutation (0.614, 95% CI 0.285–0.864 vs. 0.915, 95% CI 0.766–0.972; *p* = 0.038 in the meta-regression test). However, there was no significant correlation between CDX2 expression and mismatch repair protein and KRAS mutation.

### 3.4. Correlation between CDX2 Expression and Survival

The correlations between CDX2 expression and survivals were investigated. Patients with CDX2 expression had better overall and cancer-specific survivals than those without CDX2 expression (HR 0.735, 95% CI 0.599–0.901 and HR 0.574, 95% CI 0.431–0.764, respectively; Table 5). In the sensitivity analysis, each eligible study had no effect on pooled HR.

## 4. Discussion

CDX2, a caudal-type homeobox gene, is involved in the proliferation and differentiation of intestinal epithelial cells [45]. CDX2 is well known as a specific marker of the intestinal mucosa. In CRCs, CDX2 expression is often used as a diagnostic marker, however its capability as a prognostic marker remains unclear. Because CDX2 expression rates can vary based on the evaluation criteria employed, it is important to ascertain the impact of different evaluation criteria on these measurements. Such a comparison is not possible from a single study, and, hence, a meta-analysis such as ours becomes necessary. To the best of our knowledge, this study is the first meta-analysis on the diagnostic and prognostic roles of CDX2 immunohistochemical expression in CRCs.

A single marker with both diagnostic and prognostic capabilities, would be practically very useful. The CDX2 protein is specifically expressed in intestinal epithelial cells [8]. In an in vitro study, CDX2 knockdown promoted the proliferation of colon cancer cells [46]. Furthermore, loss of CDX2 expression correlated with an increase in mortality [8,9,10]. Olsen et al. reported no difference in CDX2 expression between normal tissue and colorectal tumors [47]. It can be difficult to determine the prognostic role of the marker when the sensitivity of the evaluation criteria is high. If the evaluation criteria is low, CDX2 expression rate is also found to be low. Therefore, the diagnostic and prognostic roles of CDX2 expression can differ based on evaluation criteria. To compare these differences, a meta-analysis is more useful than an individual study.

The evaluation criteria for CDX2 expression are unclear. We investigated the relation between CDX2 expression and the cut-off value. In our study, there was no significant difference in CDX2 expression rates across evaluation criteria and results were similar in primary and metastatic CRCs. Basically, an increase in cut-off value is expected to decrease CDX2 expression. However, our results indicate that there is no change in CDX2 expression rates with changes in evaluation criteria. In metastatic CRCs, CDX2 expression rates in the 0% and 50% cut-offs were 0.942 (95% CI 0.877–0.974) and 0.855 (95% CI 0.569–0.963), respectively. There was no significant difference in CDX2 expression rates among evaluation criteria in the meta-regression test. In the medullary carcinoma subgroup, a significant difference in CDX2 expression rates was observed between cut-offs. CDX2 expression rates were decreased by increasing the cut-off value. This result differs from the overall data. Based on our result, a lower cut-off can be appropriated to be the proper cut-off for CDX2 positivity in the diagnosis and prediction of prognosis. At the 0% cut-off, CDX2 expression was significantly correlated with better overall survival (OS) and cancer-specific survival (CSS). However, at a cut-off of 50%, the correlation for prognosis was found in CSS, but not in OS. Thus, in predicting a patient’s prognosis, a lower cut-off is more appropriate than a higher cut-off. Nonetheless, further evaluation is necessary to determine the proper cut-off value.

In the present study, we evaluated CDX2 expression rates according to histologic subtypes of CRCs. The CDX2 expression rates were similar between adenocarcinomas, mucinous carcinomas, and micropapillary carcinomas. However, the CDX2 expression rate of medullary carcinoma was significantly lower than that of other subtypes. The impact of the evaluation criteria on CDX2 expression rates in medullary carcinomas was investigated. Evaluation criteria for CDX2 expression of medullary carcinoma were 0%, 10%, and 25% in previous studies [10,28,41]. In medullary carcinoma, there was a significant difference in CDX2 expressions between the evaluation criteria in the meta-regression test (*p* = 0.003). However, because there was no difference of CDX2 expression as could be determined by evaluation criteria in overall cases, further evaluation based on subtype is needed.

Because CDX2 is known to be a specific marker for the intestinal mucosa, its diagnostic role can be useful for metastatic CRCs. CDX2 expression rates of various metastatic foci were investigated. CDX2 expression rates of metastatic CRCs in the lung, ovary, urinary bladder, and uterine cervix were 0.984, 0.896, 0.962, and 0.967, respectively. Thus, the expression rates of metastatic CRCs were higher than those of primary CRCs (0.882, 95% CI 0.774–0.861). Therefore, CDX2 may be a useful marker for the differentiation of metastatic CRCs. In previous studies, CDX2 loss or downregulation was significantly correlated with poor differentiation grade of CRCs [13,43,48]. However, in the present study, there was no significant correlation between tumor differentiation and CDX2 expression (*p* = 0.077 in the meta-regression test). As there is no correlation between differentiation and CDX2 expression, the diagnostic impacts on the differential diagnosis between poorly differentiated carcinomas are more important.

In the literature, the downregulation of CDX2 is correlated with MMR deficiency, BRAF mutations, right-sided tumors, and poor differentiation [47]. In our study, we investigated the correlation of CDX2 expression with various clinicopathological parameters, MMR deficiency, KRAS mutation, and BRAF^V600E^ mutation. There was no significant correlation between MMR deficiency, KRAS mutations and CDX2 expression (*p* = 0.066 and *p* = 0.519 in the meta-regression test, respectively). However, patients with the BRAF^V600E^ mutation had a significantly lower CDX2 expression rate than those without this mutation (*p* = 0.038 in the meta-regression test). Tomasello et al. reported the result of the prognostic role of CDX2 through a meta-analysis [49]. The researchers analyzed 16 eligible articles and included results which involved immunohistochemistry and mRNA expressions. However, unlike our study, they analyzed only the prognostic role of CDX2.

The current study has several limitations. First, in metastatic foci, the subgroup analysis based on histologic subtypes could not be performed owing to insufficient information from eligible studies. For the same reason, a detailed analysis based on evaluation criteria could not be performed in each histologic subtype.

## 5. Conclusions

In conclusion, CDX2 expression rates were high in primary and metastatic CRCs while there was no significant difference among the evaluation criteria. CDX2 can be a useful marker for differentiating between CRCs and malignant tumors of unknown origin and its expression is useful as a predictor for the prognosis of patients with CRCs.

## Figures and Tables

**Figure 1 diagnostics-12-00757-f001:**
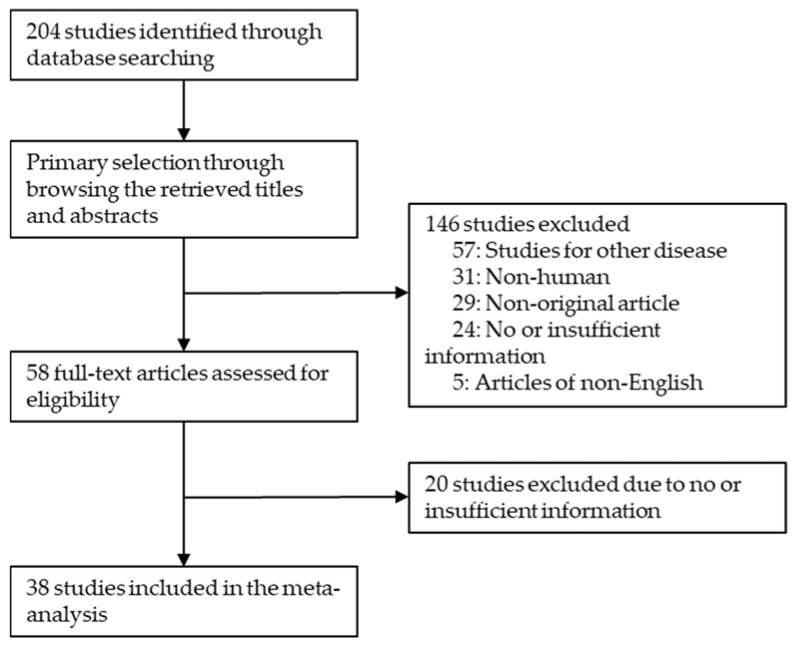
Flow chart of study search and selection methods.

**Table 1 diagnostics-12-00757-t001:** Main characteristics of eligible studies.

Author, Year	Location	AntibodyClone	Manufacturer	Criteria	Tumor	Subgroup	CDX2	
							Positive	Negative
Abouelkhair 2021	Egypt	mAB (EPR2764Y)	Cell Marque	50%	CRC		16	0
					mCRC		18	1
Asgari-Karchekani 2020	Iran	mAb	Dako	10%	CRC		53	29
Baba 2009	USA	CDX2-88	Biogenex	0%	CRC		438	183
					CRC	Medullary	47	33
Bakaris 2008	Turkey		Novacastra	50%	CRC		30	4
Barbareschi 2003	USA	CDX2-88	Biogenex	0%	CRC		58	2
					mCRC	Lung	30	0
Boulagnon-Rombi 2018	France	mAB (EPR2764Y)	Zytomed	0%	CRC		278	25
Cecchini 2019	UK	IR080	Dako	ND	CRC		187	23
		CDX2-88	Abcam					
Chu 2004	USA	CDX2-88	Biogenex	5%	CRC	Signet ring cell	8	1
Dabir 2018	Denmark	mAB (EPR2764Y)	Cell Marque	0%	mCRC		67	5
				10%			63	9
				50%			56	16
den Uil 2021	Netherlands	mAB (EPR2764Y)	Cell Marque	0%	CRC		154	192
Droy-Dupré 2015	France	CDX2-88	Biogenex	50%	CRC		84	38
Groisman 2004	Israel	CDX2-88	Biogenex	1%	mCRC	Ovary	15	0
Hamada 2017	USA	CDX2-88	Biogenex	0%	CRC		330	107
Hansen 2018	Denmark	mAb	Dako	50%	CRC	Test cohort	505	66
					CRC	Validation cohort	536	50
Hestetun 2021	Norway	mAB (EPR2764Y)	Cell Marque	50%	CRC		402	41
Hinoi 2001	USA	Developed		ND	CRC	Medullary	4	11
					CRC	WD	25	0
Inaguma 2017	USA	Clone D11D10	Cell signaling Tech.	0%	CRC		401	53
Kaimaktchiev 2004	USA	CDX2-88	Biogenex	10%	CRC		950	159
Kim 2006	Korea	CDX2-88	Biogenex	5%	CRC	Micropapillary	48	7
					CRC	Adenocarcinoma	91	28
Knösel 2012	Germany	CDX2-88	Biogenex	ND	CRC		232	168
Landau 2014	USA	CDX2-88	Biogenex	0%	CRC		178	27
Lin 2014	USA	mAB (EPR2764Y)	Cell Marque	0%	CRC	Medullary	12	6
Logani 2005	USA	CDX2-88	Biogenex	10%	mCRC	Ovary	21	1
Ma 2019	USA	CDX2-88	Biogenex	0%	CRC		451	55
Minoo 2010	USA	AMT28	Abcam	ND	CRC		324	76
Moskaluk 2003	USA	AM392	Biogenex	0%	CRC		60	0
Okoń 2004	Krakow	CDX2-88	Biogenex	0%	CRC		48	10
Panarelli 2012	USA	CDX2-88	Biogenex	50%	CRC		159	2
Pozos-Ochoa 2018	Mexico	DK-CDX2	Dako	0%	CRC	Mucinous	15	2
					CRC	Signet ring cell	6	6
Raspollini 2003	Italy	C7C/D4	Biogenex	0%	mCRC	Cervix	14	0
Roy 2012		mAb	Biocare	ND	CRC		5	0
					mCRC	Urinary bladder	12	0
Sayar 2015	Turkey	AMT28	Novocastra-Leica	0%	CRC		100	11
Sen 2015	India	mAB (EPR2764Y)	Cell Marque	10%	CRC		67	1
				50%	CRC		38	30
Shin 2010	Korea	CDX2-88	Biogenex	10%	mCRC	Ovary	30	11
Werling 2003	Brazil	CDX2-88	Biogenex	25%	CRC		74	1
Winn 2009	Rhode Island	CDX2-88	Biogenex	25%	CRC	Medullary	3	13
					CRC	PD	18	15
Winn 2010	USA	CDX2-88	Biogenex	10%	CRC	PD	33	11
					CRC	Medullary	6	9
					CRC	Signet ring cell	9	1
					CRC	WD	109	2
Zheng 2009	China		Cell signaling Tech.	ND	CRC		72	8

ND, no description; CRC, colorectal cancer; mCRC, metastatic colorectal cancer; WD, well-differentiated; PD, poorly differentiated.

**Table 2 diagnostics-12-00757-t002:** Meta-analysis for the CDX2 expression rate in the primary and metastatic colorectal carcinoma.

	Number of Subset	Fixed Effect [95% CI]	Heterogeneity Test [*p*-Value]	Random Effect [95% CI]	Egger’s Test
Primary	43	0.784 [0.774, 0.794]	<0.001	0.882 [0.774, 0.861]	0.229
Adenocarcinoma	19	0.851 [0.837, 0.863]	<0.001	0.886 [0.837, 0.923]	0.346
Mucinous carcinoma	1	0.882 [0.632, 0.970]	1.000	0.882 [0.632, 0.970]	-
Medullary carcinoma *	5	0.513 [0.427, 0.597]	0.011	0.436 [0.269, 0.618]	0.167
Micropapillary carcinoma	1	0.873 [0.756, 0.938]	1.000	0.873 [0.756, 0.938]	-
Signet ring carcinoma	3	0.690 [0.476, 0.845]	0.077	0.772 [0.406, 0.944]	0.028
0%	15	0.752 [0.735, 0.768]	<0.001	0.817 [0.731, 0.880]	0.204
5%	3	0.797 [0.731, 0.850]	0.211	0.816 [0.713, 0.888]	0.426
10%	7	0.835 [0.813, 0.854]	<0.001	0.839 [0.699, 0.922]	0.845
25%	3	0.546 [0.404, 0.682]	<0.001	0.702 [0.173, 0.963]	0.703
50%	8	0.866 [0.848, 0.881]	<0.001	0.875 [0.786, 0.930]	0.982
Metastatic	10	0.849 [0.802, 0.886]	0.016	0.893 [0.820, 0.938]	0.006
Lung	1	0.984 [0.789, 0.999]	1.000	0.984 [0.789, 0.999]	-
Ovary	3	0.791 [0.667, 0.878]	0.055	0.896 [0.605, 0.980]	0.105
Urinary bladder	1	0.962 [0.597, 0.998]	1.000	0.962 [0.597, 0.998]	-
Uterine cervix	1	0.967 [0.634, 0.998]	1.000	0.967 [0.634, 0.998]	-
0%	3	0.942 [0.877, 0.974]	0.550	0.942 [0.877, 0.974]	0.223
1%	1	0.969 [0.650, 0.998]	1.000	0.969 [0.650, 0.998]	-
10%	3	0.826 [0.747, 0.884]	0.056	0.848 [0.687, 0.934]	0.565
50%	2	0.797 [0.697, 0.870]	0.124	0.855 [0.569, 0.963]	-

CI, Confidence interval. *, Significant difference between medullary carcinoma and adenocarcinoma in the meta-regression test (*p* <0.001).

**Table 3 diagnostics-12-00757-t003:** Comparisons of CDX2 expression rates between various clinicopathological parameters.

	Number of Subset	Fixed Effect [95% CI]	Heterogeneity Test [*p*-Value]	Random Effect [95% CI]	Egger’s Test [*p*-Value]	Meta-Regression Test [*p*-Value]
Sex						
Male	6	0.799 [0.770, 0.826]	<0.001	0.871 [0.696, 0.952]	0.234	0.357
Female	6	0.720 [0.694, 0.745]	<0.001	0.780 [0.605, 0.892]	0.475	
Tumor location						
Right colon	8	0.739 [0.716, 0.761]	<0.001	0.757 [0.632, 0.850]	0.088	0.597
Left colon/Rectum	8	0.728 [0.696, 0.757]	<0.001	0.824 [0.573, 0.942]	0.032	
Histologic grade						
WD/MD	8	0.742 [0.717, 0.765]	<0.001	0.849 [0.686, 0.936]	0.354	0.077
PD	8	0.601 [0.548, 0.651]	<0.001	0.633 [0.484, 0.760]	0.544	
pT stage						
pT1/pT2	1	0.455 [0.265, 0.659]	1.000	0.455 [0.265, 0.659]	-	0.373
pT3/pT4	3	0.757 [0.730, 0.783]	<0.001	0.801 [0.427, 0.956]	0.219	
Lymph node metastasis						
Present	2	0.424 [0.348, 0.504]	0.011	0.559 [0.223, 0.848]	-	0.570
Absent	2	0.489 [0.422, 0.557]	0.009	0.821 [0.108, 0.994]	-	
Venous invasion						
Present	4	0.719 [0.655, 0.775]	<0.001	0.783 [0.545, 0.916]	0.282	0.747
Absent	4	0.787 [0.762, 0.810]	<0.001	0.833 [0.559, 0.952]	0.038	
Lymphatic invasion						
Present	1	0.897 [0.846, 0.933]	1.000	0.897 [0.846, 0.933]	-	-
Absent	1	0.888 [0.848, 0.918]	1.000	0.888 [0.848, 0.918]	-	
Perineural invasion						
Present	3	0.897 [0.838, 0.936]	0.545	0.897 [0.838, 0.936]	0.576	0.963
Absent	3	0.898 [0.881, 0.913]	0.389	0.898 [0.881, 0.913]	0.259	
pTNM stage						
Stage I and II	2	0.735 [0.690, 0.776]	<0.001	0.783 [0.436, 0.944]	-	0.937
Stage III and IV	2	0.740 [0.697, 0.780]	<0.001	0.797 [0.471, 0.946]	-	
PD-L1 expression						
Positive	2	0.716 [0.663, 0.763]	<0.001	0.634 [0.314, 0.868]	-	0.246
Negative	2	0.848 [0.812, 0.879]	<0.001	0.854 [0.555, 0.965]	-	

CI, Confidence interval; WD, well differentiated; MD, moderately differentiated.

**Table 4 diagnostics-12-00757-t004:** Comparisons of genetic mutation between colorectal cancer with and without CDX2 expression.

	Number of Subset	Fixed Effect [95% CI]	Heterogeneity Test [*p*-Value]	Random Effect [95% CI]	Egger’s Test [*p*-Value]	Meta-Regression Test [*p*-Value]
*Mismatch repair protein*						
Deficient	8	0.652 [0.614, 0.688]	<0.001	0.634 [0.486, 0.761]	0.876	0.066
Proficient	8	0.785 [0.762, 0.806]	<0.001	0.852 [0.676, 0.941]	0.335	
*KRAS* mutation						
Present	2	0.808 [0.757, 0.850]	<0.001	0.904 [0.545, 0.987]	-	0.519
Absent	2	0.728 [0.689, 0.763]	<0.001	0.810 [0.482, 0.951]	-	
*BRAF^V600E^* mutation						
Present	3	0.607 [0.536, 0.674]	<0.001	0.614 [0.285, 0.864]	0.211	0.038
Absent	3	0.845 [0.820, 0.867]	<0.001	0.915 [0.766, 0.972]	0.244	

CI, Confidence interval.

**Table 5 diagnostics-12-00757-t005:** Comparisons of prognosis between colorectal cancer with and without CDX2 expression.

	Number of Subset	Fixed Effect [95% CI]	Heterogeneity Test [*p*-Value]	Random Effect [95% CI]	Egger’s Test [*p*-Value]
*Overall survival*	4	0.735 [0.599, 0.901]	0.606	0.735 [0.599, 0.901]	0.387
*Cancer-specific survival*	5	0.592 [0.472, 0.743]	0.237	0.574 [0.431, 0.764]	0.216

CI, Confidence interval.

## Data Availability

No new data were created or analyzed in this study. Data sharing is not applicable to this article.

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
