# Peer review of "Diagnostic and Prognostic Roles of CDX2 Immunohistochemical Expression in Colorectal Cancers"

_diagnostics, 2022, doi:10.3390/diagnostics12030757_

Round 1

Reviewer 1 Report

Gist/Review:  A review of this sort where meta-analysis is carried out on CDX2  needs substantial discussions.  Although the authors attempt to bring clarity on diagnostics and prognostics of this, I would have personally appreciated had the authors made some decent attempt in analysing the protein interaction networks, pathways from these datasets.  This will be a  good way.

All the statistical interpretations, plots were deemed to be okay. 

Third, . the authors would have overcome the non-finding of histological subtypes, had they used operands ( AND, OR and NOT) in PubMed searches Minor but essential

I think Figure 1 would come in methods NOT results

a phenolyzer map would be a nice addition for this review.  Pl draw that figure

In intro, the CRC is abbreviated twice, pl remove the second such instance

The abstract needs to be rewritten for brevity.  For example

The study IS aimed....

Line 86.  Pl use plural   random effect MODELS

Line 212:  carcinomas   ARE  ( replace with is

Author Response

Review 1.

Gist/Review:  A review of this sort where meta-analysis is carried out on CDX2 needs substantial discussions. Although the authors attempt to bring clarity on diagnostics and prognostics of this, I would have personally appreciated had the authors made some decent attempt in analysing the protein interaction networks, pathways from these datasets. This will be a good way.

All the statistical interpretations, plots were deemed to be okay. 

Third, the authors would have overcome the non-finding of histological subtypes, had they used operands (AND, OR and NOT) in PubMed searches Minor but essential

Response:

                  We corrected the information for the operands.

I think Figure 1 would come in methods NOT results

Response:

                  As a recommendation, we corrected the location of Fig.1.

a phenolyzer map would be a nice addition for this review. Pl draw that figure

Response:

In our knowledge, a phenolyzer is a software that implements phenotype-based prioritization of candidate genes for human diseases. However, we analyzed the protein expression of only CDX2.

In intro, the CRC is abbreviated twice, pl remove the second such instance

Response:

                  As a recommendation, we corrected it.

The abstract needs to be rewritten for brevity. For example

Response:

                  As a recommendation, we corrected the abstract.

The study IS aimed....

Response:

                  As a recommendation, we corrected the sentence.

Line 86.  Pl use plural   random effect MODELS

Response:

                  As a recommendation, we replaced.

Line 212:  carcinomas ARE ( replace with is

Response:

                  As a recommendation, we replaced “is” to “are”.

Reviewer 2 Report

It's a well written systematic review to evaluate the diagnostic and prognostic role of the immunohistochemical expression of the single biological marker  CDX2 in colorectal cancers. Overall this topic has been extensively studied and there are numerous publications available, however, the authors present a comprehensive overview and meta-analysis that summarizes the available data. Of potential interest to the pathologists, surgical and medical oncologists, molecular biologists interested in CRC biology and prognostification.

Author Response

Review 2.

It's a well written systematic review to evaluate the diagnostic and prognostic role of the immunohistochemical expression of the single biological marker CDX2 in colorectal cancers. Overall this topic has been extensively studied and there are numerous publications available, however, the authors present a comprehensive overview and meta-analysis that summarizes the available data. Of potential interest to the pathologists, surgical and medical oncologists, molecular biologists interested in CRC biology and prognostification.

Response:

                  Thank you for the careful review.